# Effect of Magnesium Supplementation on Circulating Biomarkers of Cardiovascular Disease

**DOI:** 10.3390/nu12061697

**Published:** 2020-06-06

**Authors:** Alvaro Alonso, Lin Y. Chen, Kyle D. Rudser, Faye L. Norby, Mary R. Rooney, Pamela L. Lutsey

**Affiliations:** 1Department of Epidemiology, Rollins School of Public Health, Emory University, Atlanta, GA 30322, USA; 2Cardiovascular Division, Department of Medicine, University of Minnesota, Minneapolis, MN 55455, USA; chenx484@umn.edu; 3Division of Biostatistics, School of Public Health, University of Minnesota, Minneapolis, MN 55455, USA; rudser@umn.edu; 4Division of Epidemiology and Community Health, School of Public Health, University of Minnesota, Minneapolis, MN 55454, USA; flopez@umn.edu (F.L.N.); lutsey@umn.edu (P.L.L.); 5Department of Epidemiology, Johns Hopkins Bloomberg School of Public Health, Baltimore, MD 21287, USA; mroone12@jhu.edu

**Keywords:** magnesium, proteomics, randomized trial

## Abstract

(1) Background: Magnesium supplementation may be effective for the prevention of cardiometabolic diseases, but the mechanisms are unclear. Proteomic approaches can assist in identifying the underlying mechanisms. (2) Methods: We collected repeated blood samples from 52 individuals enrolled in a double-blind trial which randomized participants 1:1 to oral magnesium supplementation (400 mg magnesium/day in the form of magnesium oxide) or a matching placebo for 10 weeks. Plasma levels of 91 proteins were measured at baseline with follow-up samples using the Olink Cardiovascular Disease III proximity extension assay panel and were modeled as arbitrary units in a log_2_ scale. We evaluated the effect of oral magnesium supplementation for changes in protein levels and the baseline association between serum magnesium and protein levels. The Holm procedure was used to adjust for multiple comparisons. (3) Results: Participants were 73% women, 94% white, and had a mean age of 62. Changes in proteins did not significantly differ between the two intervention groups after correction for multiple comparisons. The most statistically significant effects were on myoglobin [difference −0.319 log_2_ units, 95% confidence interval (CI) (−0.550, −0.088), *p =* 0.008], tartrate-resistant acid phosphatase type 5 (−0.187, (−0.328, −0.045), *p =* 0.011), tumor necrosis factor ligand superfamily member 13B (−0.181, (−0.332, −0.031), *p =* 0.019), ST2 protein (−0.198, (−0.363, −0.032), *p =* 0.020), and interleukin-1 receptor type 1 (−0.144, (−0.273, −0.015), *p =* 0.029). Similarly, none of the associations of baseline serum magnesium with protein levels were significant after correction for multiple comparisons. (4) Conclusions: Although we did not identify statistically significant effects of oral magnesium supplementation in this relatively small study, this study demonstrates the value of proteomic approaches for the investigation of mechanisms underlying the beneficial effects of magnesium supplementation. Clinical Trials Registration: ClinicalTrials.gov NCT02837328.

## 1. Introduction

Mounting evidence suggests that moderately elevated concentrations of circulating magnesium may reduce the risk of coronary heart disease and atrial fibrillation. This evidence comes from prospective observational studies [1,2], Mendelian randomization studies [3,4], and studies of magnesium supplementation in secondary prevention [5], even though observational studies do not support an effect of dietary magnesium on cardiovascular disease [6]. Mechanisms underlying this potential protective effect are unclear, but may include antiarrhythmic effects, improved glucose homeostasis, better vascular tone and endothelial function, and reduced oxidative stress and inflammation [1,7,8,9].

Recent advances in the field of proteomics allow the efficient evaluation of multiple proteins in biological tissues. This provides an opportunity to assess simultaneous multiple markers of distinct mechanistic pathways [10]; however, this approach has rarely been applied to the study of the effects of oral magnesium supplementation [11].

To provide novel insights into the pathways linking magnesium and cardiovascular risk, we evaluated the effect of oral magnesium supplementation on multiple cardiovascular-related circulating proteins measured using a novel proteomic assay. This analysis was done using repeated blood samples collected from 52 participants in a double-blind randomized trial testing efficacy and tolerability of 400 mg/day of magnesium oxide compared to placebo for the prevention of supraventricular arrhythmias.

## 2. Methods

### 2.1. Study Population

Between March and June 2017, we recruited and randomized 59 men and women to receive 400 mg/daily of oral magnesium in the form of magnesium oxide or a matching placebo for 12 weeks to determine the effect of oral magnesium supplementation on supraventricular arrhythmias (ClinicalTrials.gov #NCT02837328). Details about recruitment, inclusion and exclusion criteria, and study procedures have been published elsewhere [12]. Briefly, we included men and women 55 years of age or older without a prior history of heart disease (coronary heart disease, heart failure, atrial fibrillation), stroke, or kidney disease, not using magnesium supplements, and living in the Minneapolis/St. Paul, MN area. Eligible participants attended a baseline visit where they underwent a basic physical exam, blood collection, and had a heart rhythm monitor applied (Zio^®^ XT, iRhythm Technologies, Inc., San Francisco, CA, USA). After wearing the monitor for two weeks, participants were randomized 1:1 to 400 mg/daily of magnesium or a matching placebo and the study intervention was mailed. Twelve weeks after the baseline exam (10 weeks after starting study intervention), participants underwent a follow-up visit, which included blood collection. For this analysis, we included 52 trial participants with blood samples available for proteomic analysis at baseline (pre-randomization) and the follow-up visit. The University of Minnesota Institutional Review Board approved the study protocol and all participants provided written informed consent.

### 2.2. Intervention

The University of Minnesota Institute for Therapeutics Discovery and Drug Development manufactured the study intervention (400 mg of magnesium in the form of magnesium oxide capsules) and the placebo (lactose) following Good Manufacturing Practices. The University of Minnesota Investigational Drug Service managed bottling. Participants and study staff were blinded to the treatment given. Compliance with the intervention was excellent. As previously reported, the magnesium group participants took 75% of tablets, whereas those in the placebo group took 83.4%, based on pill count. During the course of the trial, 50% of the participants who were assigned to magnesium and 7% who were assigned to the placebo commented on gastrointestinal changes at any point in the study, but only one participant in the magnesium arm discontinued the blinded study treatment [12].

### 2.3. Blood Biomarker Analysis

Participants were asked to fast for eight hours prior to blood draws at the baseline and follow-up visits. Serum and plasma samples were obtained and processed using standard procedures and stored in −80 °C freezers. Circulating magnesium was measured in serum samples using the Roche Cobas 6000 at the University of Minnesota Advanced Research and Diagnostic Laboratory.

### 2.4. Proteomic Measurements

Relative levels of 92 proteins were measured in never-thawed plasma samples using the Olink Cardiovascular III panel (www.olink.com, Olink Proteomics, Uppsala, Sweden). The Olink panel uses a proximity extension assay (PEA) to measure multiple protein biomarkers simultaneously [13]. Briefly, for each protein, a unique pair of oligonucleotide-labeled antibody probes bind to the targeted protein and if the two probes are brought into close proximity, the oligonucleotides will hybridize in a pairwise manner. The addition of a DNA polymerase leads to a proximity-dependent DNA polymerization event, generating a unique polymerase chain reaction target sequence. The resulting DNA sequence is subsequently detected and quantified using a microfluidic real-time polymerase chain reaction instrument (Biomark HD, Fluidigm, South San Francisco, CA, USA). Data are then quality controlled and normalized using an internal extension control and an interplate control to adjust for intra- and inter-run variation. The protein levels are given in Normalized Protein eXpression (NPX) units, which is an arbitrary measure on the log_2_-scale, with higher values corresponding to higher protein concentrations. All assay characteristics, including detection limits and measurements of assay performance and validations, are available from the manufacturer’s webpage (http://www.olink.com). The analyses were based on 1 μL of plasma for each panel of 92 assays. To avoid batch effects, samples from the two intervention groups and the two visits were randomized across assay plates. Each plate included internal controls, as described previously, to adjust for technical variation and sample irregularities [13]. Due to technical issues, one of the protein assays (C-C motif chemokine 22) was not performed, resulting in measurements of 91 proteins.

### 2.5. Other Covariates

At the baseline and follow-up clinic visits, participants self-reported their age, sex, race, and smoking status. Trained technicians measured height, weight and blood pressure, and performed a phlebotomy. Anthropometric measures were obtained with the participant wearing light clothing and no shoes. Blood pressure was measured three times with the participant sitting after a five-minute rest.

### 2.6. Statistical Analysis

The primary goal of the analysis was to evaluate the effect of magnesium supplementation versus placebo for change in levels of multiple cardiovascular-related circulating proteins. Of the 91 measured proteins in the array, we excluded those with >25% values below the limit of detection across both groups combined as well as those with excessive within-person variability, for which an intervention effect would be unlikely to be detected. One protein [spondin-1 (SPON1)] was excluded due to a large number of values below the limit of detection. To evaluate within-person variability, we determined pairwise correlations between measurements from samples collected at the baseline and follow-up visits in the placebo group and excluded proteins with r <0.3. Three proteins were identified as having excessive variability and were subsequently excluded: ephrin type-B receptor 4 (EPHB4), azurocidin (AZU1), and kallikrein-6 (KLK6). Appendix A presents complete results for the pairwise correlations and the proportion of samples with values below the limit of detection. All 87 proteins were available for analysis.

We used multiple linear regression with robust variance estimation to evaluate the effect of oral magnesium supplementation on change in levels of individual proteins (modeled as log_2_-transformed units). The dependent variable was the difference in protein levels (follow-up visit minus baseline visit). Models adjusted for randomization stratification factor (age <65 vs. ≥65) and baseline value of the protein. Since this was an exploratory hypothesis-free analysis, multiple comparisons were taken into account using the Holm procedure [14]. A secondary analysis was performed adjusting for sex. In an additional analysis, we assessed the baseline cross-sectional associations of serum magnesium with individual proteins considering baseline levels of the protein as the dependent variable and serum magnesium, modeled as a continuous variable, as the main independent variable, adjusting for age (continuous), sex, and race. The analyses were conducted using SAS version 9.4 (SAS Inc., Cary, NC, USA).

The sample size of the original trial (*n* = 60) was determined to detect a difference in the change of ectopic supraventricular beats (primary endpoint) between treatment groups of 0.79 standard deviation units with 80% power and 5% type I two-sided error and assuming that five participants would not complete the follow-up.

## 3. Results

Of 59 participants in the trial, 52 provided samples at baseline and follow-up visit and had available proteomic data. Of these, 24 were assigned to the magnesium intervention and 28 to the placebo group (Figure 1). The mean age of the two groups was similar (62 years), but the proportion of women was higher in the magnesium intervention group: 88% versus 61% in the placebo group (Table 1). Change in magnesium concentration was significantly higher for those assigned to magnesium supplementation compared to placebo (0.035 mmol/L, 95% confidence interval 0.015, 0.06, *p* = 0.003). This magnitude of change is equivalent to 0.6 standard deviations of baseline magnesium concentration.

An analysis of pairwise correlations between baseline protein levels showed most proteins were not strongly correlated to each other with three clusters, including a total of eleven proteins, correlated with r >0.8 (Figure 2). The first cluster included P-selectin (SELP), bleomycin (BLM) hydrolase, junctional adhesion molecule A (JAMA), caspase-3 (CASP3), platelet-derived growth factor (PDGF) subunit A, and platelet endothelial cell adhesion molecule (PECAM1). The second cluster included tumor necrosis factor receptor 1 (TNFR1), tumor necrosis factor receptor 2 (TNFR2), and interleukin-18-binding protein (IL18BP). Finally, the third cluster included carboxypeptidase A1 (CPA1) and carboxypeptidase B (CPB1).

The effect of oral magnesium supplementation versus placebo on 87 circulating proteins is reported in Figure 3 and Appendix A. None of the associations were statistically significant after accounting for multiple comparisons with the Holm procedure. The strongest effect was on levels of myoglobin, with a difference of −0.319 NPX units (95% confidence interval −0.550, −0.088; *p* = 0.008) in the change over time between the intervention and placebo groups. Table 2 and Appendix A present results for the five proteins with between-group differences with *p*-value <0.05. Associations were of similar magnitude after adjustment for sex (Appendix A). 

When evaluating the association of serum magnesium with circulating protein levels, we did not identify any statistically significant associations using the Holm procedure to account for multiple comparisons (Figure 4 and Appendix A). Four proteins had associations with a *p*-value <0.05 (Table 3). The strongest was the association between serum magnesium and epidermal growth factor receptor (beta = 0.053 NPX units, 95% CI 0.013, 0.093, *p* = 0.011, per 0.04 mmol/L difference in serum magnesium).

## 4. Discussion

In this analysis, we evaluated the effect of oral magnesium supplementation on the circulating levels of multiple proteins related to cardiovascular disease. We observed that, compared to placebo, oral magnesium supplementation led to changes in levels of several proteins. For those with *p*-values <0.05, all associations were in the hypothesized direction, with Mg supplementation versus placebo associated with more advantageous levels of cardiovascular proteins. Similarly, we observed associations between baseline serum magnesium and several circulating proteins. However, none of the associations explored were statistically significant after correcting for multiple comparisons.

Growing observational evidence indicates that lower levels of circulating magnesium are associated with increased risk of atrial fibrillation and coronary heart disease [1,2]. In addition, experimental studies show that magnesium supplementation can be effective in the secondary prevention of cardiac arrhythmias [5,15]. Mechanisms underlying these associations, however, are unknown. Though we failed to identify significant effects of magnesium on circulating proteins, the magnitude of the protein changes after a relatively short intervention supports the use of proteomic panels in future larger studies of magnesium supplementation. These panels will facilitate the identification of biomarkers and physiological pathways responsible for the potential effects of magnesium on cardiovascular risk.

To date, the use of proteomic approaches to evaluate effects of magnesium supplementation has been extremely limited. In a crossover trial of 14 healthy overweight individuals, supplementation with 500 mg/day of magnesium (in the form of magnesium citrate) vs. placebo for 4 weeks did not result in consistent changes in circulating inflammatory biomarkers. However, urine proteomic profiling identified significant differences in the expression profiles of the proteome, but not specific proteins [11]. Similarly, a study of 52 overweight and obese individuals randomized to 350 mg/day of magnesium or placebo for 24 weeks evaluated the effect of the intervention on multiple circulating biomarkers of inflammation and endothelial dysfunction. No significant differences were reported between the two intervention groups [16]. 

The proteins, for which we observed some evidence of effect (albeit not significant after multiple correction), are involved in muscle structure and oxygen storage (myoglobin) [17], immune function and inflammation (tumor necrosis factor ligand superfamily member 13B [18], ST2 protein [19], interleukin-1 receptor type 1) [20], and bone metabolism (tartrate-resistant acid phosphatase type 5 [21]). Of interest, higher circulating levels of ST2 have been linked to adverse cardiovascular outcomes [22]. Similarly, in our cross-sectional analysis, higher concentrations of serum magnesium were associated with higher levels of proteins involved in multiple functions (epidermal growth factor receptor) [23], cell adhesion (junctional adhesion molecule A [24], platelet endothelial cell adhesion molecule [25]), and oxidative stress protection (paraoxonase 3) [26]. These effects were consistent with some of the proposed effects of magnesium supplementation, including reductions in oxidative stress and inflammation [7,8].

Our study had some strengths, including the randomized design, the demonstrated efficacy of the intervention in increasing circulating magnesium [12,27], and the simultaneous assessment of multiple circulating proteins. However, this analysis was hindered by the limited sample size and absence of replication in an independent sample. The limited sample size precluded studying specifically participants with hypomagnesemia. Also, we lacked information on kidney function, which influences levels of numerous proteins. However, by including a healthy sample, this was less likely to be an issue. Finally, we did not collect data on dietary magnesium and, among female participants, menopausal status, use of hormone therapy, or circulating estrogens. We are uncertain of the potential effect of these variables on our effect estimates. However, the randomized design would contribute to balancing them across the control and intervention groups. 

## 5. Conclusions

In summary, our study demonstrated the potential value of proteomic approaches for the investigation of mechanisms underlying the beneficial effects of magnesium supplementation. Future trials in larger samples are needed to establish with certainty the physiological impact of magnesium and, therefore, inform the development of magnesium-based interventions for the prevention of cardiovascular and metabolic diseases.

## Figures and Tables

**Figure 1 nutrients-12-01697-f001:**
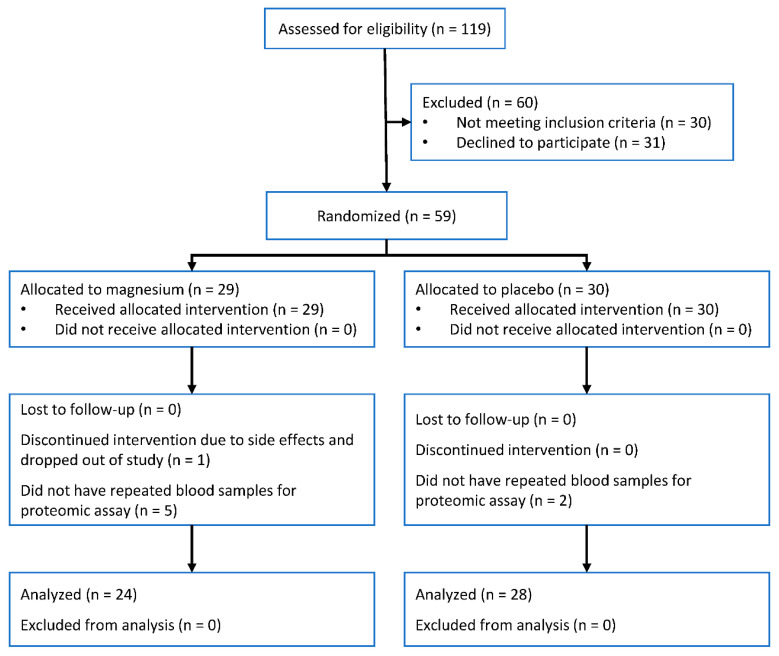
Participant flow diagram.

**Figure 2 nutrients-12-01697-f002:**
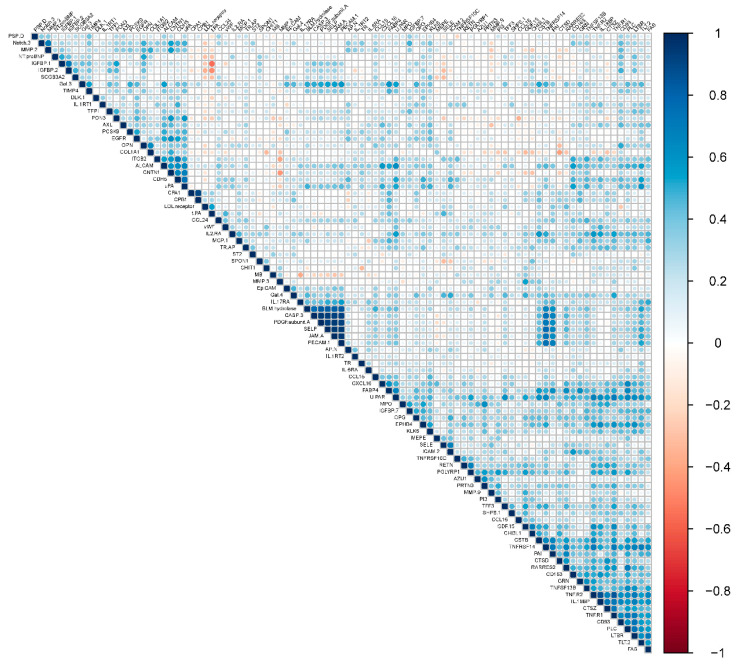
Pairwise correlations between baseline levels of individual proteins.

**Figure 3 nutrients-12-01697-f003:**
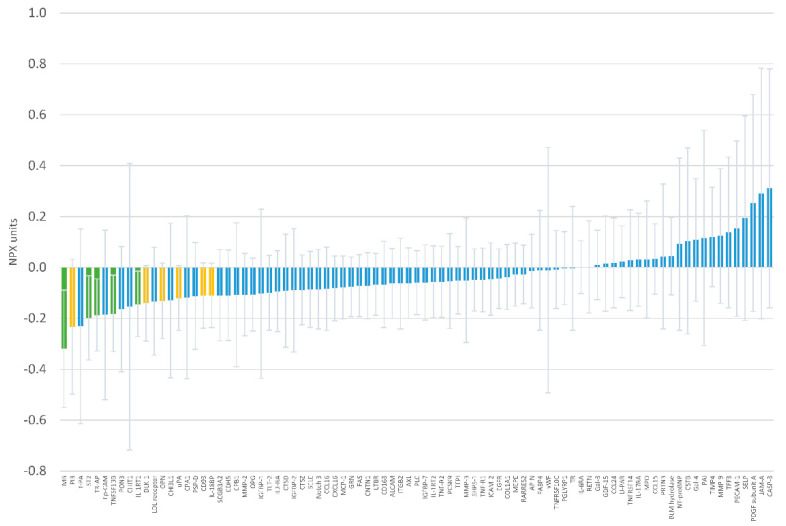
Mean difference in 12-week change of individual protein levels in Normalized Protein eXpression (NPX) units comparing oral magnesium supplementation to placebo. Error bars correspond to 95% confidence. Green bars indicate differences with *p* <0.05, yellow bars with *p*-value ≥0.05 and <0.10.

**Figure 4 nutrients-12-01697-f004:**
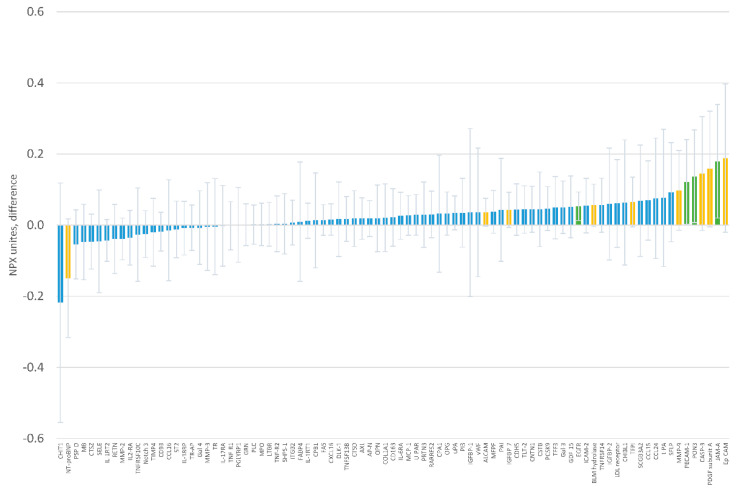
Baseline association of serum magnesium with individual protein levels in Normalized Protein eXpression (NPX) units. Coefficients correspond to the difference in protein levels per 0.04 mmol/L difference in serum magnesium. Error bars correspond to 95% confidence intervals. Green bars indicate differences with *p* <0.05, yellow bars with *p*-value ≥0.05 and <0.10.

**Table 1 nutrients-12-01697-t001:** Baseline characteristics of study participants by treatment assignment. Values presented are mean (SD) or frequency (%) where indicated.

	Magnesium (400 mg Daily)	Placebo
*N*	24	28
Age, years	62 (5)	62 (6)
Women, *n* (%)	21 (88)	17 (61)
Non-white, *n* (%)	2 (8)	1 (4)
Body mass index, kg/m^2^	28.3 (5.1)	27.8 (4.2)
Systolic blood pressure, mmHg	118 (15)	119 (17)
Diastolic blood pressure, mmHg	73 (8)	71 (8)
Serum magnesium, mmol/L	0.86 (0.06)	0.84 (0.05)
Hypomagnesemia, *n* (%) *	2 (8.3)	2 (7.1)

* Hypomagnesemia defined as circulating magnesium <0.75 mmol/L.

**Table 2 nutrients-12-01697-t002:** Effect of magnesium supplementation on selected circulating proteins, expressed as a difference in change between magnesium and placebo group, in Normalized Protein eXpression (NPX) units. Results for effects with *p*-value <0.05.

Protein		Difference in Change in Protein (NPX Units)	95% CI	*p*-Value
MB	Myoglobin	−0.319	−0.550, −0.088	0.008
TR-AP	Tartrate-resistant acid phosphatase type 5	−0.187	−0.328, −0.045	0.011
TNFSF13B	Tumor necrosis factor ligand superfamily member 13B	−0.181	−0.332, −0.031	0.019
ST2	ST2 protein	−0.198	−0.363, −0.032	0.020
IL-1RT1	Interleukin-1 receptor type 1	−0.144	−0.273, −0.015	0.029

Results from the general linear model with a difference in protein values between follow-up and baseline measurements in Normalized Protein eXpression (NPX) units as the dependent variable, treatment assignment as the main independent variable, adjusted for baseline levels of the protein and age stratum.

**Table 3 nutrients-12-01697-t003:** Association of baseline serum magnesium with levels of selected circulating proteins. Estimates correspond to a difference in protein levels, expressed in Normalized Protein eXpression (NPX) units, per 0.04 mmol/L difference in serum magnesium. Results for associations with *p*-value <0.05.

Protein		Difference in Protein Levels (NPX Units)	95% CI	*p*-Value
EGFR	Epidermal growth factor receptor	0.053	0.013, 0.093	0.011
JAM-A	Junctional adhesion molecule A	0.180	0.020, 0.339	0.028
PON3	Paraoxonase 3	0.137	0.039, 0.007	0.039
PECAM-1	Platelet endothelial cell adhesion molecule	0.121	0.002, 0.241	0.046

Results from general linear model with baseline levels of protein in Normalized Protein eXpression (NPX) units as the dependent variable, serum magnesium as the main independent variable, adjusted for age, sex, and race.

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
