# Peer review of "Effect of Magnesium Supplementation on Circulating Biomarkers of Cardiovascular Disease"

_nutrients, 2020, doi:10.3390/nu12061697_

Round 1

Reviewer 1 Report

This study investigated the effect of magnesium oxide supplementation (400 mg/day) on cardiovascular parameters in an experimental group of 26 patients as compared to an equivalent group of patients receiving placebo. Magnesium was administered for 12 weeks, at which point various cardiovascular and hematic parameters were assessed vs. a pre-established baseline 14 week before. The results provided in this study indicate that no significant differences were observed between control group and experimental group once the proper corrections for multiple comparisons were put in place. The most statistically significant effects were observed on myoglobin, tartrate-resistant acid phosphatase type 5, tumor necrosis factor ligand superfamily member 13B, ST2 protein, and interleukin-1 receptor type 1. No meaningful correlation of protein content vs. basal magnesium level were observed. 

The conclusion of the authors is that, although no statistically significant effects of oral magnesium supplementation were observed in this relatively small study, the results support the value of proteomic approaches for the investigation of mechanisms underlying the beneficial effects of magnesium supplementation.

Comments:

The study appears to be properly conducted, and the conclusions are supported by the data presented here. 

A few, however, need further clarification. 

  1. why the authors selected 12 weeks of treatment when data in the literature (referenced in the study) showed that 24 weeks of treatment with 350 mg MgOx/day showed no significant difference?
  2. why was MgOx oxide selected when its bioavailability is significantly lower than many other magnesium salts (e.g. Mg-citrate, Mg-aspartate, etc.).
  3. Women represented a large number of the experimental group. Given the age of the participants, it is reasonable to anticipate that the majority, if not the totality of them, were in menopause. As the levels of estrogens have cardiovascular protective effects, which percentage of the women enrolled in the two groups were in menopause? Which percentage of the women were taking estrogen replacement? Were the levels of estrogens tested? None of these criteria were presented as an exclusion parameter, nor were they addressed in the discussion section. 

Author Response

Response to Reviewer 1

“This study investigated the effect of magnesium oxide supplementation (400 mg/day) on cardiovascular parameters in an experimental group of 26 patients as compared to an equivalent group of patients receiving placebo. Magnesium was administered for 12 weeks, at which point various cardiovascular and hematic parameters were assessed vs. a pre-established baseline 14 week before. The results provided in this study indicate that no significant differences were observed between control group and experimental group once the proper corrections for multiple comparisons were put in place. The most statistically significant effects were observed on myoglobin, tartrate-resistant acid phosphatase type 5, tumor necrosis factor ligand superfamily member 13B, ST2 protein, and interleukin-1 receptor type 1. No meaningful correlation of protein content vs. basal magnesium level were observed.

The conclusion of the authors is that, although no statistically significant effects of oral magnesium supplementation were observed in this relatively small study, the results support the value of proteomic approaches for the investigation of mechanisms underlying the beneficial effects of magnesium supplementation.

Comments:

The study appears to be properly conducted, and the conclusions are supported by the data presented here.”

We thank the reviewer for the positive comments.

“A few, however, need further clarification.

Why the authors selected 12 weeks of treatment when data in the literature (referenced in the study) showed that 24 weeks of treatment with 350 mg MgOx/day showed no significant difference?”

The decision to intervene for a 12-week period was based on a meta-analysis of randomized trials of magnesium supplementation that showed that extending the trial for longer than 12 weeks would lead to small changes beyond those observed in the first 12 weeks (Fig 3B in Zhang et al, ref 25 in the manuscript.) As we reported in our previous publication, this intervention was effective in increasing circulating magnesium (difference in intervention vs placebo: 0.07 mEq/L, 95%CI 0.03-0.12, p = 0.002; Lutsey et al, ref. 10 in the manuscript.)

“Why was MgOx oxide selected when its bioavailability is significantly lower than many other magnesium salts (e.g. Mg-citrate, Mg-aspartate, etc.).”

The decision to use magnesium oxide (MgO) rather than organic salts, such as magnesium citrate or magnesium aspartate, was of convenience, while maintaining scientific rigor. MgO, an inorganic compound, is the formulation commonly used in multivitamins and trial of magnesium supplementation. As the reviewer mentions, organic compounds, such as magnesium citrate, have been suggested to be more bioavailable, but their molecular weight is relatively large (C6H5MgO7: 213 g/mol), and therefore it would have required intake of three large softgels per day to achieve our desired dosage of 400 mg daily (magnesium aspartate has an even bigger molecular weight). Magnesium oxide is a much smaller molecule and required only one capsule daily to reach 400 mg. Even using magnesium oxide, our trial demonstrated an effect of the intervention on circulating magnesium.

“Women represented a large number of the experimental group. Given the age of the participants, it is reasonable to anticipate that the majority, if not the totality of them, were in menopause. As the levels of estrogens have cardiovascular protective effects, which percentage of the women enrolled in the two groups were in menopause? Which percentage of the women were taking estrogen replacement? Were the levels of estrogens tested? None of these criteria were presented as an exclusion parameter, nor were they addressed in the discussion section.”

The reviewer makes an excellent point. Unfortunately, we did not collect information on menopausal status or use of hormone therapy, or measured circulating estrogens. However, the randomized design would have reduced the risk of any of these variables confounding our results. We have added this as a limitation to the manuscript.

Changes to the manuscript

We have added the following text to the discussion (lines 225-228):

“Finally, we did not collect data on dietary magnesium and, among female participants, menopausal status, use of hormone therapy, or circulating estrogens. We are uncertain of the potential effect of these variables on our effect estimates. The randomized design, though, would have contributed to balance them across control and intervention groups.”

Reviewer 2 Report

In this paper, plasma levels of 91 proteins were measured in baseline and follow-up samples using 27 the Olink Cardiovascular Disease III proximity extension assay panel after magnesium supplementation. The manuscript is well written and conclusions are supported by the analysis performed. However, the results are not very informative and I have several concerns about the utility of this study for the potential readers of Nutrients.

Author Response

Thank you for the feedback on our manuscript. A point-by-point response to the comments is provided in the attached document.

Reviewer 3 Report

This study evaluated the effect of oral magnesium supplementation on cardiovascular risk through a proteomic assay is novel. It provides a potential to understand the mechanism of oral magnesium on cardiovascular diseases protection. 

I do not have much to say about the method, as I'm not familiar with the proteomic assay myself. But there are possibly other issues to be considered and discussed:

1. As presented in Table 1, the average magnesium level is normal in both intervention and control groups, so how the extra magnesium supplements will and to what extent to have actual effect in protecting the cardiovascular risk is unknown. Will these extra magnesium just flow away from the blood stream?

2. How about dietary intake or dietary magnesium intake? Have you adjusted in the model? 

3. How about the medication use for these participants? Because magnesium will interact with some medications such as antibiotics. So maybe it is good to adjust these potential medication use in the model. Even though those participants are free of CVD, they may likely to have used medication (e.g. antibiotics) from time to time, given their age group. 

There are other minor issues may also be considered: 1. I cannot see any figures besides the supplemental one, so make sure you have uploaded them. 2. Tables, can you remove the outlines of the table to meet the publication standards for tables?

Author Response

Response to Reviewer 3

“This study evaluated the effect of oral magnesium supplementation on cardiovascular risk through a proteomic assay is novel. It provides a potential to understand the mechanism of oral magnesium on cardiovascular diseases protection.

I do not have much to say about the method, as I'm not familiar with the proteomic assay myself. But there are possibly other issues to be considered and discussed:

  1. As presented in Table 1, the average magnesium level is normal in both intervention and control groups, so how the extra magnesium supplements will and to what extent to have actual effect in protecting the cardiovascular risk is unknown. Will these extra magnesium just flow away from the blood stream?”

Though the mean magnesium concentration was in the normal range, we have previously reported that in this population 7% had concentrations below the threshold for clinical deficiency (<1.82 mg/dL; <1.5 mEq/L) and 38% below the threshold for subclinical deficiency (<2.07 mg/dL; <1.7 mEq/L) (Lutsey et al, ref. 10 in the manuscript). In addition, we demonstrated that supplementary magnesium had an effect on circulating magnesium equivalent to 0.7 standard deviations of circulating magnesium, which could be translated in beneficial health outcomes based on the epidemiologic literature exploring the associations between circulating magnesium and cardiovascular disease risk in large cohorts. For example, a meta-analysis of prospective studies found that 0.2 mmol/L (approximately 0.48 mg/dl) increment in circulating magnesium was associated with a 30% lower risk of cardiovascular disease (del Gobbo et al, ref. 1 in the manuscript). Even if we are uncertain of the exact mechanisms through which circulating magnesium (and magnesium supplementation) affects cardiovascular risk, this type of intervention offers promise of being effective for cardiovascular prevention. Studies like the current one, exploring proteomic changes with supplementation, have the potential to assist in understanding mechanisms.

“2. How about dietary intake or dietary magnesium intake? Have you adjusted in the model?”

In this trial, we did not collect information on diet and, therefore, we lack information on dietary magnesium. Because of the randomized design, however, the two groups would tend to be balanced on average dietary magnesium between the intervention and control groups. Also, effects of dietary magnesium on circulating proteins are likely to be mediated through the impact they have on circulating magnesium. Finally, we excluded participants taking oral magnesium supplements. We clarify in the discussion that dietary information was not available.

Changes to the manuscript

We have added the following text to the discussion (lines 225-228):

“Finally, we did not collect data on dietary magnesium and, among female participants, menopausal status, use of hormone therapy, or circulating estrogens. We are uncertain of the potential effect of these variables on our effect estimates. The randomized design, though, would have contributed to balance them across control and intervention groups.”

“3. How about the medication use for these participants? Because magnesium will interact with some medications such as antibiotics. So maybe it is good to adjust these potential medication use in the model. Even though those participants are free of CVD, they may likely to have used medication (e.g. antibiotics) from time to time, given their age group.”

We did ask participants for current use of medication at baseline. Not unexpectedly, a majority of participants were using at least one medication (mostly antihypertensives, statins, antidiabetics, anti-inflammatories, or antidepressants). Antibiotics were not particularly prevalent, since they are used only for short periods of time. We did not adjust for medication use in our analysis because prevalence of individual medications was very low, any differences across intervention and control group would tend to be balanced by randomization, and the small sample size precluded inclusion of an excessive number of covariates in the model.

“There are other minor issues may also be considered: 1. I cannot see any figures besides the supplemental one, so make sure you have uploaded them. 2. Tables, can you remove the outlines of the table to meet the publication standards for tables?”

We will be sure that figures are available when we resubmit the manuscript and have removed side outlines from the tables.

Reviewer 4 Report

The present study is a further evaluation of  data gathered within a "pilot study" on (the same) 59 subjects published by the same authors in NUTRIENTS 2018 (citation No  10). Concerning the (present) study population the reader is referred to the foregoing paper where he learns that exclusion criteria were "prior history of heart disease" and that 6.9% of the study population were hypomagnesemic and 37.9% suboptimal. In contrast, serum-Mg concentrations are summarzed by 2.1 mg% in the present paper!(Table 1;data should be presented in mmol/L)! - Side effects mentioned in 2018 are now omitted! - - On the other hand, the data presented in the present paper are of general interest. However important clinical data must be included. Discussion of the present measured data in the hypomagnesemic subgroup would be of special interest

Author Response

Response to Reviewer 4

“The present study is a further evaluation of data gathered within a "pilot study" on (the same) 59 subjects published by the same authors in NUTRIENTS 2018 (citation No  10). Concerning the (present) study population the reader is referred to the foregoing paper where he learns that exclusion criteria were "prior history of heart disease" and that 6.9% of the study population were hypomagnesemic and 37.9% suboptimal. In contrast, serum-Mg concentrations are summarzed by 2.1 mg% in the present paper!(Table 1;data should be presented in mmol/L)! - Side effects mentioned in 2018 are now omitted! - - On the other hand, the data presented in the present paper are of general interest. However important clinical data must be included. Discussion of the present measured data in the hypomagnesemic subgroup would be of special interest.”

In response to the reviewer’s comment, we now provide some additional details about the original trial in the methods, results, and table 1.

Ideally, we would conduct stratified analyses by circulating magnesium concentrations. Unfortunately, given the limited sample size, restricting analyses to the hypomagnesemic group would be very limited. We mention this as a limitation and recommendation for future work.

Changes to the manuscript

We have clarified exclusion criteria (lines 71-72):

“we included men and women 55 years of age or older, without a prior history of heart disease (coronary heart disease, heart failure, atrial fibrillation), stroke, or kidney disease”

We have added information on compliance with the intervention and associated adverse effects (lines 87-91):

“Compliance with the intervention was excellent, as previously reported: the in the magnesium group participants took 75% of tablets, whereas those in the placebo group took 83.4%, based on pill count. During the course of the trial, 50% of participants assigned to magnesium and 7% assigned to placebo commented on gastrointestinal changes at any point in the study, but only one participant, in the magnesium arm, discontinued blinded study treatment.”

In the Discussion, we mention the limited sample size to explore specific effects in those with hypomagnesemia as a limitation (lines 222-223):

“The limited sample size precluded studying specifically participants with hypomagnesemia.”

Finally, we have added the proportion of participants that were hypomagnesemic, as well as the concentrations of serum magnesium in mmol/L (Table 1).

Round 2

Reviewer 3 Report

The manuscript has been improved.

Author Response

We thank the reviewer for his/her feedback.

Reviewer 4 Report

I agree with the authors´ comments. Please report Mg concentrations exclusively in mmol/L (line 153; Table 1)

Author Response

We now use mmol/L to present magnesium concentrations throughout the manuscript.